

# Preparation of mitochondria to measure superoxide flashes in angiosperm flowers

Chulan Zhang[1,*], Fengshuo Sun[2,*], Biao Xiong[3] and Zhixiang Zhang[1]

[1] College of Nature Conservation, Beijing Forestry University, Beijing, China
[2] College of Biological Sciences and Biotechnology, Beijing Forestry University, Beijing, China
[3] College of Tea Science, Guizhou University, Guizhou Province, China
[*] These authors contributed equally to this work.

Corresponding author
Zhixiang Zhang, zxzhang@bjfu.edu.cn

## ABSTRACT

**Background**. Mitochondria are the center of energy metabolism and the production of reactive oxygen species (ROS). ROS production results in a burst of "superoxide flashes", which is always accompanied by depolarization of mitochondrial membrane potential. Superoxide flashes have only been studied in the model plant *Arabidopsis thaliana* using a complex method to isolate mitochondria. In this study, we present an efficient, easier method to isolate functional mitochondria from floral tissues to measure superoxide flashes.

**Method**. We used 0.5 g samples to isolate mitochondria within <1.5 h from flowers of two non-transgenic plants (*Magnolia denudata* and *Nelumbo nucifera*) to measure superoxide flashes. Superoxide flashes were visualized by the pH-insensitive indicator MitoSOX Red, while the mitochondrial membrane potential ($\Delta\Psi$ m) was labelled with TMRM.

**Results**. Mitochondria isolated using our method showed a high respiration ratio. Our results indicate that the location of ROS and mitochondria was in a good coincidence. Increased ROS together with a higher frequency of superoxide flashes was found in mitochondria isolated from the flower pistil. Furthermore, a higher rate of depolarization of the $\Delta\Psi$ m was observed in the pistil. Taken together, these results demonstrate that the frequency of superoxide flashes is closely related to depolarization of the $\Delta\Psi$ m in petals and pistils of flowers.

## INTRODUCTION

Mitochondria are widely distributed organelles in eukaryotic cells where they perform important roles generating energy, regulating physiological activities, and maintaining cellular metabolism (*Hatefi, 1985*; *Yang, Mukda & Chen, 2018*). The major role of mitochondria is the generation of ATP by oxidative phosphorylation through the electron transport chain (*Hatefi, 1985*). In addition to energy production, mitochondria are also the center of reactive oxygen species (ROS) production in organisms under biotic or abiotic stress (*Paital & Chainy, 2014*; *Yang et al., 2016*). The isolation of mitochondria has deepened research on metabolism and stress in plants (*Day, Neuburger & Douce, 1985*). In

1985, mitochondria from 300 g of pea leaves were isolated and purified by centrifugation on a Percoll gradient containing a linear gradient of polyvinylpyrrolidone-25 (0–10%, w/v) to obtain only 20 mg mitochondrial protein (*Day, Neuburger & Douce, 1985*). After that, mitochondria were isolated from *Arabidopsis thaliana* using differential centrifugation and further purified using a continuous colloidal density gradient (*Lyu et al., 2018*; *Sweetlove, Taylor & Leaver, 2007*). In addition, crude isolation of mitochondria in leaves using density gradient centrifugation revealed higher respiratory coupling than that observed in purified mitochondria *Keech, Dizengremel & Gardestrom, 2005*. It is well known that mitochondria must be purified to extract mitochondrial DNA and the proteome (*Ahmed & Fu, 2015*; *Kim et al., 2015*), but the time required and the sampling method were not suitable in many mitochondrial studies, particularly in non-green tissues such as flowers. Crude isolation of the intact and functional mitochondria is crucial for the measurement of superoxide flashes in plants.

Superoxide flashes are 10-s events that occur spontaneously and suddenly in mitochondria and reflect electrical and chemical activities (*Feng et al., 2017*). Superoxide flashes were first defined as transient events of the mitochondrial matrix-targeted biosensor mt-cp YFP (*Wei & Dirksen, 2012*). As mt-cp YFP is sensitive to pH, superoxide flashes can be visualized by chemical probes, including ROS indicators, such as MitoSOX for superoxide and 2.7-dichlorodihyfrofluorescein (DCF) for $H_2O_2$ (*Feng et al., 2017*; *Zhang et al., 2013*). Interestingly, cp-YFP superoxide flashes are correlated with depolarization of the mitochondrial membrane potential ($\Delta\Psi_m$) (*Zhang et al., 2013*). Previous studies have shown that ROS modulate a variety of physiological events, including growth, stress, thermogenesis, and diseases (*Jastroch, 2017*; *Keunen et al., 2015*; *Kuznetsov et al., 2017*; *Maksimov et al., 2018*; *Yang et al., 2016*). It is clear that the accumulation of ROS are closely associated with superoxide flashes. In animals, superoxide flashes and ROS bursts are involved in various physiological activities, such as oxidative stress, metabolism, and aging (*Pouvreau, 2010*; *Wei et al., 2011*). Thus, there is a close relationship between superoxide flashes and mitochondrial energy metabolism. Considering the importance of the mitochondrial respiratory chain and energy metabolism, it is of great significance to study mitochondrial superoxide flashes in plants.

Superoxide flashes have been well studied in cells and isolated mitochondria of animals, and the cp YFP-flash signals are always associated with the loss of $\Delta\Psi_m$ (labeled with TMRM) (*Li et al., 2012*). Superoxide flashes observed with the chemical probes MitoSOX and DCF reveal the same results and frequency as cp YFP flashes (*Zhang et al., 2013*). In plant tissues, superoxide flashes have only been studied in the roots of Arabidopsis and the cp-YFP signals changed with different respiratory substrates (*Schwarzlander et al., 2011*), but no study has explored superoxide flashes in other non-transgenic tissues of plants. Floral tissues in plants are important organs involved in various physical activities, including thermogenesis, pollination, and reproduction (*Luo et al., 2010*; *Thien et al., 2009*). Mitochondrial energy metabolism and oxygen consumption are closely related to floral thermogenesis and reproduction (*Miller et al., 2011*); thus, it is necessary to combine the activity of mitochondrial superoxide flashes with a study of floral reproduction in plants. As isolating plant mitochondria using a previous method was likely to influence

mitochondrial viability and the mitochondrial-targeted cp-YFP is hardly expressed in xylophyta flowers, a suitable method to study superoxide flashes in floral tissues is crucial.

To address these issues, some important modifications were devised based on previous methods to study superoxide flashes (*Zhang et al., 2013*). We developed an efficient method to isolate high viability mitochondria in floral tissues of *Magnolia denudata* and *Nelumbo nucifera*. As these are non-transgenic flowers, superoxide flashes were first visualized by loading the plants with MitoSOX Red, while the $\Delta\Psi_m$ was labelled with TMRM. These methods facilitated study of mitochondrial energy metabolism and physiological activities in non-transgenic flowers of angiosperms. This quick and sample-saving protocol greatly improved the viability of mitochondria and efficiency of the experiment of superoxide flashes in non-green plant tissues.

## MATERIALS & METHODS

### Plant materials/plant growth

*M. denudata* was grown on the campus of Beijing Forestry University ($40°00'02''$ N, $116°20'15''$, a.s.l., 60 m). Pistils and petals of 15 flowers were collected during afternoons in March and April. *N. nucifera* was grown in Bajia Country Park ($40°00'50''$N, $116°19'39''$E, a.s.l., 47 m). Receptacles and petals of nearly 10 flowers were collected during afternoons in July–August.

### Solutions

Method A: Grinding buffer: 0.3 M sucrose, 25 mM $Na_4P_2O_4$, 2 mM EDTA, 10 mM $KH_2PO_4$, 1% (w/v) polyvinylpyrrolidone-40, 1% (w/v) defatted bovine serum albumin (BSA), 4 mM cysteine, and 20 mM ascorbic acid were added just prior to grinding. pH was adjusted to 7.5 with KOH. Resuspension buffer: 0.3 M sucrose, 10 mM N-Tris [hydroxymethyl]-methyl-2-aminoethanesulfonic acid (TES-KOH), and 0.1% BSA, pH = 7.5. Mitochondrial basic incubation medium: 0.3 M sucrose, 10 mM TES-KOH. 10 mM NaCl, 5 mM $KH_2PO_4$, 2 mM $MgSO_4$, and 0.1% BSA, pH = 7.2.

Method B: Grinding buffer: 0.3 M sucrose, 25 mM $Na_4P_2O_4$, 2 mM EDTA, 10 mM $KH_2PO_4$, 1% (w/v) polyvinylpyrrolidone-40, 1% (w/v) defatted bovine serum albumin (BSA) and 20 mM ascorbic acid were added just prior to grinding. pH was adjusted to 7.5 with KOH. Resuspension buffer: 0.3 M sucrose and 10mM TES-KOH, pH = 7.5. Preparation of a single linear PVP-40 gradient in 28% (v/v) Percoll: 0.3 M sucros, 10 mM $KH_2PO_4$, 0.1% BSA, 28% (v/v) Percoll and a linear gradient of 0–10% (w/v) PVP-40 (top to bottom) in a 30 ml centrifuge tube. pH = 7.2. Mitochondrial basic incubation medium: 0.3 M sucrose, 10 mM TES-KOH. 10 mM NaCl, 5 mM $KH_2PO_4$, 2 mM $MgSO_4$, and 0.1% BSA, pH = 7.2.

### Isolation of mitochondria

Method A: Our efficiency method to obtain crude, high viability mitochondria. All steps were carried at 4 °C on ice. Mitochondria of magnolia were isolated from style and petal tissues while mitochondria of lotus were isolated from receptacle and petal tissues. About 0.5 g of pistil or petal tissues were cut up from each species into 1 $mm^3$-fragments with

scissors. They were ground in 1–2 ml of grinding buffer using a pestle with a small amount of quartz. The extract was filtered through 20 μm nylon mesh and then centrifuged at 2,000× g for 10 min to remove most of the thylakoid membranes and intact chloroplasts. The supernatant was transferred to a new tube and centrifuged at 12,000× g for 20 min. The pellet was resuspended in 1 ml resuspension buffer and centrifuged for 5 min at 1,500× g to remove the residual intact chloroplasts. This new supernatant was centrifuged for 20 min at 12,000× g to yield the crude mitochondria. The crude mitochondria were suspended in mitochondrial basic incubation medium and placed on ice for further studies.

Method B: to obtain purified mitochondria as previous study. All steps were carried at 4 °C on ice. Mitochondria were isolated from style of magnolia and receptacle of lotus. About 45 g of pistil tissues were cut up with scissors (see above). Isolation of mitochondria was based on the method of *Sweetlove, Taylor & Leaver (2007)* with minor modification. Briefly, pistil tissues were blended in 200 ml grinding buffer (see above), filtered through 20 μm nylon mesh and then centrifuged at 1,100× g for 10 min. The supernatant was centrifuged at 18,000× g for 20 min. The pellet was resuspended in 30 ml resuspension buffer and centrifuged for 10 min at 1,100× g. This new supernatant was centrifuged for another 20 min at 18,000× g. The final pellet was suspended in 1 ml resuspension buffer and layered over the linear PVP-40 gradient in 28% (v/v) Percoll (see above), and centrifuged for 45 min at 40,000 × g. The mitochondria were found in a tight white band near the bottom of tube. The mitochondria fraction was carefully removed and resuspended in 20 ml resuspension buffer, the suspension was centrifuged for 20 min at 15,000 × g. The purified mitochondria were suspended in mitochondrial basic incubation medium and place on ice for further studies.

## Mitochondrial respiratory function assay

The oxygen consumption rates of mitochondria were determined with a Clark-type oxygen electrode (Strathkelvin 782 2-Channel Oxygen System version 1.0; Strathkelvin Instruments, Motherwell, UK) at 25 °C. A 10 μl aliquot of mitochondrial suspension was blended in 1 ml of mitochondrial basic incubation medium. The oxygen sensor signal was recorded on a computer at intervals of 0.5 s with Strathkelvin Instruments software (782 System version 1.0). Oxygen consumption was measured with 250 μM ADP (state 3) and with 5 mM succinate (state 4). The respiratory control ratio (RCR) was calculated as the ratio of state 3 to state 4 respiration. The mitochondrial suspensions with higher than a state 3 RCR were used in subsequent studies.

## Confocal imaging of Mito-ROS and $\Delta\Psi_m$

To visualize the superoxide flashes and $\Delta\Psi_m$, isolated mitochondria from pistil and petal tissues of magnolia and lotus were immobilized on round glass cover slides (pretreatment with 0.2 mg/ml poly-L-lysine for 1 h; Sigma, St. Louis, MO, USA) by centrifugation at 2,000× g for 5 min at 4 °C and mounted on an inverted microscope (Zeiss LSM 710; Carl Zeiss, Oberkochen, Germany) for imaging. To measure the subcellular locations of mitochondria and ROS, mitochondria were first incubated with 100 nM MitoTracker Green (Invitrogen, Carlsbad, CA, USA) for 30 min at 25 °C and washed in mitochondrial basic
incubation medium, then loaded with 2.5 µM MitoSOX Red for 5 min. MitoTracker Green was excited with 488 nm and emissions were collected at 500–530 nm, while MitoSOX Red was excited with 543 nm and collected at an emission wavelength of 560–620 nm. Isolated mitochondria were labelled with 2.5 µM MitoSOX Red and 5 mM succinate as a respiration substrate to measure superoxide flashes. To understand the MitoSOX-flashes behavior in the change of respiration state and uncoupler, 0.25 mM ADP and 5µM FCCP (Carbonyl cyanide 4-(trifluoromethoxy) phenylhydrazone) was added to observe the superoxide flashes. Isolated mitochondria were loaded with 50 nM TMRM and 5 mM succinate for 1 min at 25 °C to measure the $\Delta\Psi_m$. The excitation wavelength for TMRM was 543 and the emission wavelength was 550–620 nm. A total of 100 frames of $512 \times 512$ pixels were collected for a typical time-series recording. The frame rate was 50–60 frames/min. All experiments were performed at room temperature (24–26 °C).

## Data analysis

The images obtained by laser scanning confocal microscopy were analyzed using Image J 1.48v (Wayne Rasband, National Institutes of Health, Bethesda, MD, USA). Superoxide flashes and variations in the $\Delta\Psi_m$ were identified using FlashSniper (*Li et al., 2012*), and their morphological, properties, and duration were measured automatically. Statistical analyses were performed using SPSS Statistics 23.0 software (IBM Corp., Armonk, NY, USA). Images were processed and assembled using Adobe Photoshop CS 5 (Adobe Systems Corp., San Jose, CA, USA).

# RESULTS

## Respiratory function and viability of isolated mitochondria

Crude mitochondria were sampled from petal and style tissues of magnolia as shown in Fig. 1A, while mitochondria from petal and receptacle tissues of lotus were sampled as shown in Fig. 1F. A signal with excitation at 488 nm was confirmed to avoid the disturbing auto-fluorescence of intact chloroplasts. As shown in Fig. 1B and Fig. 1G, no intact chloroplasts were detected in the crude isolated mitochondria. To compare the previous method (method B) (*Day, Neuburger & Douce, 1985*) and our efficient method (method A) to isolate mitochondria, the respiratory function of the isolated mitochondria was determined with a Clark-type oxygen electrode. As a result, the RCR did not change significantly in mitochondria isolated from flowers using method A ($n = 6$), but RCR declined in isolated mitochondria using method B ($n = 6$) (Table 1). As the viability of mitochondria is reflected by the $\Delta\Psi_m$, crude isolated mitochondria were loaded with the TMRM indicator. Highly viable and highly dense mitochondria were observed in mitochondria of magnolia (Figs. 1C, 1D) and lotus (Figs. 1H , 1I). The viability of mitochondria using method B was lower than that of method A (Figs. 1E, 1J). We assessed the time consumed, amount of sample consumed, and the viability of both methods. Using method B, mitochondria were processed in $5.28 \pm 0.23$ h and consumed $43.92 \pm 3.78$ g of flower tissues ($n = 6$), whereas mitochondria were isolated within $1.13 \pm 0.14$ h with only $0.47 \pm 0.12$ g tissues ($n = 6$) using our method A. This result shows that our mitochondrial

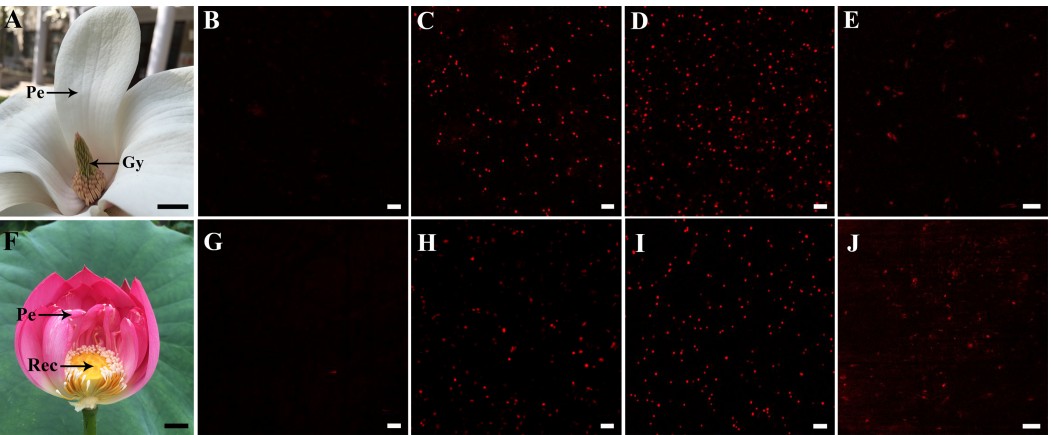

**Figure 1 Sampling and detection of isolated mitochondria.** Sampling of *Magnolia denudata* (A) and *Nelumbo nucifera* (F). Detection of auto-fluorescence of intact chloroplasts in *M. denudata* (B) and *N. nucifera* (G). Viability of isolated mitochondria in petal (C) and style (D) of *M. denudata*, petal (H) and receptacle (I) of *N. nucifera* using our efficiency method A (crude isolated mitochondria). Viability of mitochondria isolated from style in *M. denudata* (E) and from receptacle in *N. nucifera* (J) using method B (density gradient-purified mitochondria). Pe: petal, Gy: gynoecium, Rec: receptacle. Scale bar: 5 μm.(All photoes were taken by Chulan Zhang).

**Table 1 Respiratory function of isolated mitochondria with our method A and previous method B.**

| Groups | State 3 nmol O min$^{-1}$ mg$^{-1}$ | State 4 nmol O min$^{-1}$ mg$^{-1}$ | RCR |
|---|---|---|---|
| Stigma (*M. denudata*) | 269.68 ± 28.49 | 60.08 ± 6.13 | 4.49 ± 0.20[a] |
| Petal (*M. denudata*) | 276.06 ± 31.50 | 64.46 ± 7.88 | 4.29 ± 0.16[ab] |
| Receptacle (*N. nucifera*) | 257.73 ± 34.91 | 60.00 ± 8.59 | 4.30 ± 0.21[ab] |
| Petal (*N.nucifera*) | 259.14 ± 33.82 | 61.99 ± 8.68 | 4.19 ± 0.25[b] |
| Method B | 243.90 ± 35.01 | 61.89 ± 8.39 | 3.94 ± 0.18[c] |

**Notes.**
Values are mean ± S.D., $n = 6$.

isolation method was highly efficient to obtain highly viable mitochondria in the flower species.

## ROS production in floral mitochondria

To identify the intracellular site of ROS production, mitochondrial ROS were loaded with MitoSOX Red for 5 min (2.5 μM), while mitochondria were loaded with Mito Tracker Green for 30 min (100 nM). ROS production and mitochondrial location were coincident in the mitochondria isolated from petals and styles of magnolia, suggesting that mitochondria are the primary site of ROS production in this species (Figs. 2C, 2G). The same results were found in the mitochondria isolated from receptacle and petal of lotus (Figs. 2J, 2N).

The fluorescent level of ROS increased significantly in the mitochondria isolated from style compared to the petal of magnolia (Figs. 2B, 2D, 2F) ($n = 100$). In addition, similar results were found in the isolated mitochondria of lotus, as the ROS level was significantly higher in the receptacle than in the petal (Figs. 2I, 2K, 2M) ($n = 100$). Our results confirm

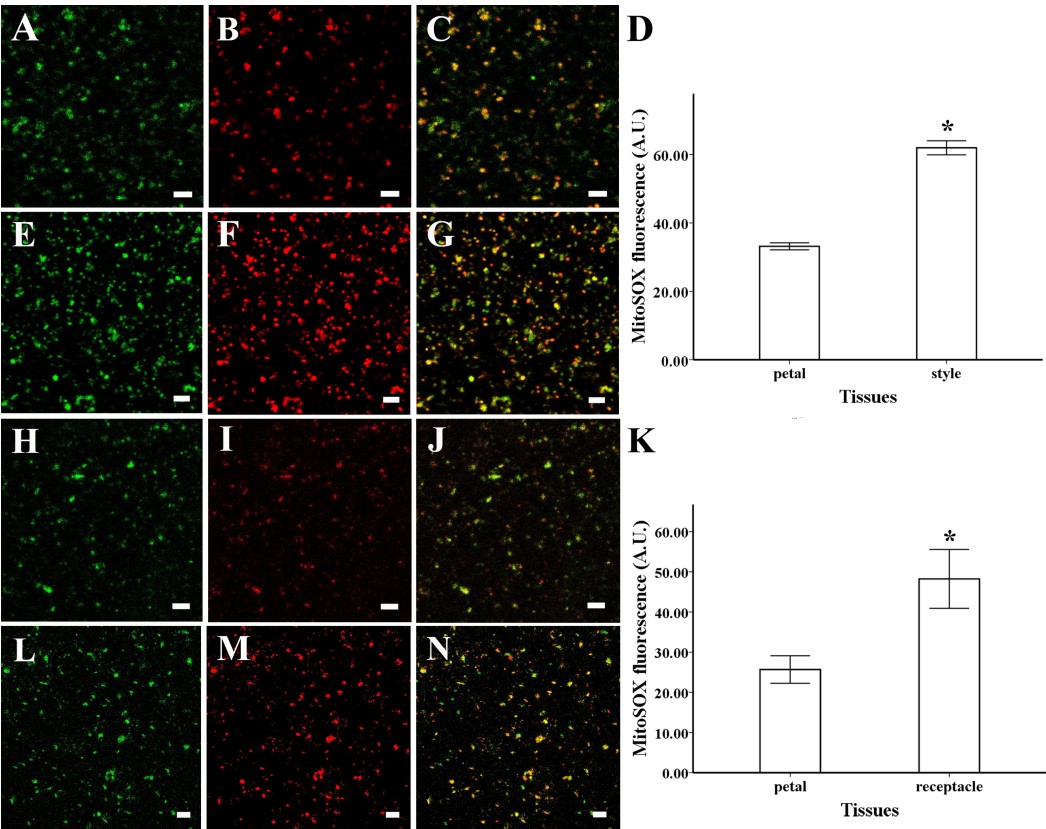

**Figure 2  ROS production and colocalization with mitochondria.** ROS production and colocalization with mitochondria isolated from *M. denudata*: mitochondria isolated from petal (A) and style (E) were visualized by Mito Tracker Green (in green color), mitochondrial ROS in mitochondria isolated from petal (B) and mitochondria isolated from style (F) was visualized by MitoSOX red (in red color), colocalization of ROS and mitochondria in petal (C) and style (G). (D) Comparison of ROS fluorescent intensity in petal and style of *M. denudata*. ROS production and colocalization with mitochondria isolated from *N. nucifera*: mitochondria isolated from petal (H) and receptacle (L) were visualized by Mito Tracker Green (in green color), mitochondrial ROS in mitochondria isolated from petal (I) and mitochondria isolated from style (M) was visualized by MitoSOX red (in red color), colocalization of ROS and mitochondria in petal (J) and receptacle (N). K. Comparison of ROS fluorescent intensity in petal and receptacle of *N. nucifera*. Scale bar: 5 μm.

that mitochondrial ROS tended to accumulate in the pistil of both magnolia and lotus, indicating that mitochondrial ROS might be more involved in the electron transport chain in the pistil than in the petal.

## Superoxide flashes in flowers

To investigate the nature of superoxide flashes in magnolia and lotus, isolated mitochondria were loaded with the ROS fluorescent probe MitoSOX Red with 5 mM succinate added as respiratory substrate. According to a previous study (*Wang et al., 2016b*), we defined the variation of fluorescence at $df/F_0 > 0.2$ within 10 s as a single superoxide flash event. A transient increase in MitoSOX fluorescence and variations in the trace were observed during 100 s in single mitochondrial events (Fig. 3A, B, and Video S1). Among these

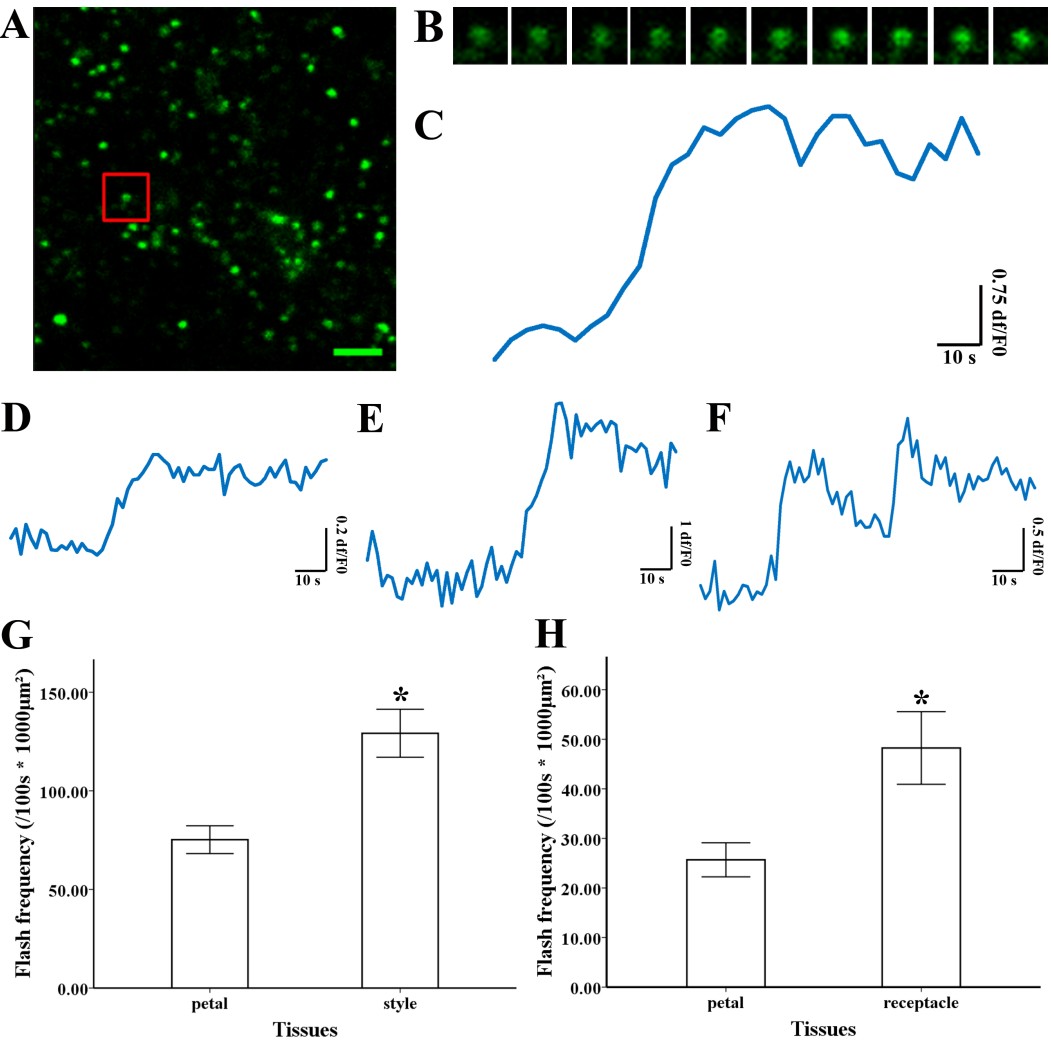

**Figure 3 Superoxide flashes visualized by MitoSOX and flashes frequency.** (A) Isolated mitochondria labeled with MitoSOX red. (B) Time-lapse images and (C) typical trace of superoxide flashes visualized by MitoSOX red. Different types of traces of superoxide flashes (D) low variation slope traces, (E) high variation slope traces, (F) multi-event traces. (G) Comparison of superoxide flashes frequency in mitochondria isolated from petal and style of *M. denudata*. (H) Comparison of superoxide flashes frequency in mitochondria isolated from petal and receptacle of *N. nucifera*. Scale bar: 5 μm.

instantaneous traces, three types of mitochondrial superoxide traces were classified (Figs. 3D–3F): low variation slope traces ($0.2 < df/F_0 < 0.5$) (Fig. 3D), high variation slope traces ($0.5 \leq df/F_0$) (Fig. 3E), and multi-event traces ($0.2 < df/F_0$) (Fig. 3F). We also compared the frequency of superoxide flashes ($/100\,s \times 1,000\,\mu m^2$) in mitochondria isolated from petals and pistils of magnolia and lotus. Notably, superoxide oxide flashes labelled with MitoSOX Red were detected at a rate of $129.18 \pm 20.11$ ($/100\,s \times 1,000\,\mu m^2$, $n = 13$) in mitochondria isolated from the magnolia style (Fig. 3G) which was significantly higher than mitochondria in the petal ($75.23 \pm 10.48/100\,s \times 1,000\,\mu m^2$, $n = 11$). In lotus (Fig. 3H), the rate of superoxide flashes was $48.24 \pm 10.24$ ($/100\,s \times 1,000\,\mu m^2$, $n = 10$) in mitochondria

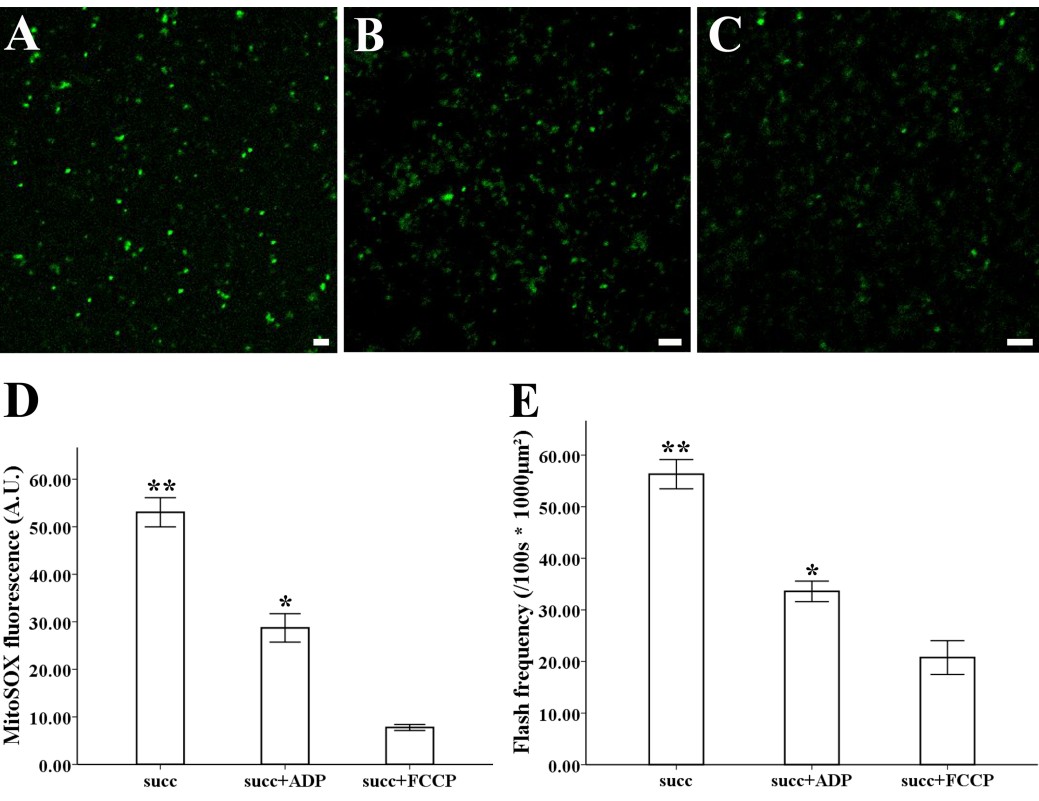

**Figure 4  ROS production and superoxide flashes in different respiratory substrate.** Isolated mitochondria from style of *M. denudata* labeled with MitoSOX red (A) 5 mM succinate, (B) 5 mM succinate and 250 μM ADP, (C) 5 mM succinate and 5 μM FCCP. Comparison of ROS production and superoxide flashes frequency in mitochondria isolated from style of *M. denudata* in different respiratory substrate: (D) Comparison of mitochondrial ROS fluorescent intensity in different respiratory substrate. (E) Comparison of mitochondrial superoxide flashes frequency in different respiratory substrate. succ: succinate. Scale bar: 5 μm.

isolated from the style, which was also significantly higher than mitochondria in the petal ($25.68 \pm 4.79$ /100 s $\times 1{,}000$ μm$^2$, $n = 10$). These results indicate that superoxide flashes, together with ROS bursts, are highly autonomous and predominantly reflect the properties and physical activities of mitochondria in different tissues and species.

The MitoSOX-flashes were closely linked to functional ETC and respiratory activity. To observe the mitochondrial behavior in the change of respiration state and uncoupler, 0.25 mM ADP and 5 μM FCCP was added for the measurement of mitochondria isolated from style of Magnolia. The addition of ADP resulted in a significantly decrease in MitoSOX fluorescence (Figs. 4B, 4D) ($n = 100$). Mitochondrial respiratory uncoupled by FCCP also led to the significantly decrease in MitoSOX signal (Figs. 4C, 4D) ($n = 100$). Similar behavior was found in the measurement of MitoSOX-flashes, the rate of superoxide flashes with succinate was significantly higher than mitochondria in the addition of ADP and FCCP (Fig. 4E) ($n = 6$). Low flashes occurred in mitochondria upon uncoupling suggested that the electrical transmembrane gradient might modulate the superoxide flashes.

## Depolarization of the mitochondrial membrane potential in flowers

To study variations in the $\Delta\Psi_m$, isolated mitochondria were labelled with TMRM, and 5 mM of succinate was added. The decline in fluorescent intensity at df/$F_0$ < −0.2 was defined as an event. Transient depolarization of the $\Delta\Psi m$ accompanied by later polarization occurred in a single mitochondrion (Fig. 5A, B and Video S2). According to the wide variation in $\Delta\Psi m$, the trace $\Delta\Psi m$ was catalogued into three types (Figs. 5D–5F): Instantaneous loss of $\Delta\Psi m$ along with instant recovery (Fig. 5D), instantaneous loss of $\Delta\Psi m$ with a short period of stability before recovery (Fig. 5E), and multi-event trace including the above two types (Fig. 5F). The frequency of a TMRM-event in mitochondria isolated from magnolia petals (544.92 $\pm$ 56.98/100 s $\times$1,000 $\mu m^2$, $n = 15$) was significantly lower than the values in the style (1,009.10 $\pm$130.10 /100 s $\times$1,000 $\mu m^2$, $n = 15$) (Fig. 5G). The same result was found in lotus (Fig. 5H) that the frequency of TMRM events in mitochondria isolated from the lotus petal was 51.94 $\pm$ 10.57 (/100 s $\times$1,000 $\mu m^2$, $n = 10$) which was lower than that in the receptacle (119.99 $\pm$ 19.00/100 s $\times$1,000 $\mu m^2$, $n = 10$). We conclude that transient and spontaneous depolarization of $\Delta\Psi m$ occurred in all tissues and the higher frequency of variation of $\Delta\Psi m$ in the pistils of flowers suggest that they have a higher level of mitochondrial dynamics.

## DISCUSSION

Isolating mitochondria from plant tissues is complex and inefficient. In a previous study, the sucrose-based differential centrifugation method requires high-speed centrifugation (40,000$\times$ g) and 300 g of sample within 5 h to obtain purified mitochondria (*Day, Neuburger & Douce, 1985*). Isolating mitochondria using the colloidal density gradient method consumes 60 g of sample and more than 4 h (*Sweetlove, Taylor & Leaver, 2007*). These methods are time- and sample-consuming, which may hinder the function and respiratory coupling of the mitochondria. In our method, we used only 0.5 g of floral tissues to obtain crude functional mitochondria in less than 1.5 h after centrifugation at a low speed ($\leq 12,000\times$ g), which only required a standard laboratory centrifuge. A previous study reported that isolating crude mitochondria from leaves results in a higher RCR than when isolating purified mitochondria, which was consistent with our results *Keech, Dizengremel & Gardestrom, 2005*. Since high mitochondrial respiration control ratio was prerequisite for superoxide flashes to occur (*Schwarzlander et al., 2011*), isolation of high viability mitochondria was necessary for the measurement of superoxide flashes. In the present study, we provide an effective and simple method to obtain highly viable mitochondria in different flower tissues to measure mitochondrial ROS, superoxide flashes and the $\Delta\Psi_m$.

Mitochondrial ROS modulate various physiological events, including stress, growth, and cell death (*Dickinson & Chang, 2011*; *NavaneethaKrishnan, Rosales & Lee, 2018*; *Sundaresan et al., 1995*). Colocalization of ROS and mitochondria in polar growing pollen tubes reveals the production of $H_2O_2$ in mitochondria during pollen germination (*Maksimov et al., 2018*). Also, ROS are produced in mitochondria until the full flower bloom stage (*Chakrabarty, Chatterjee & Datta, 2007*; *Rogers, 2012*). Our study found good coincidence

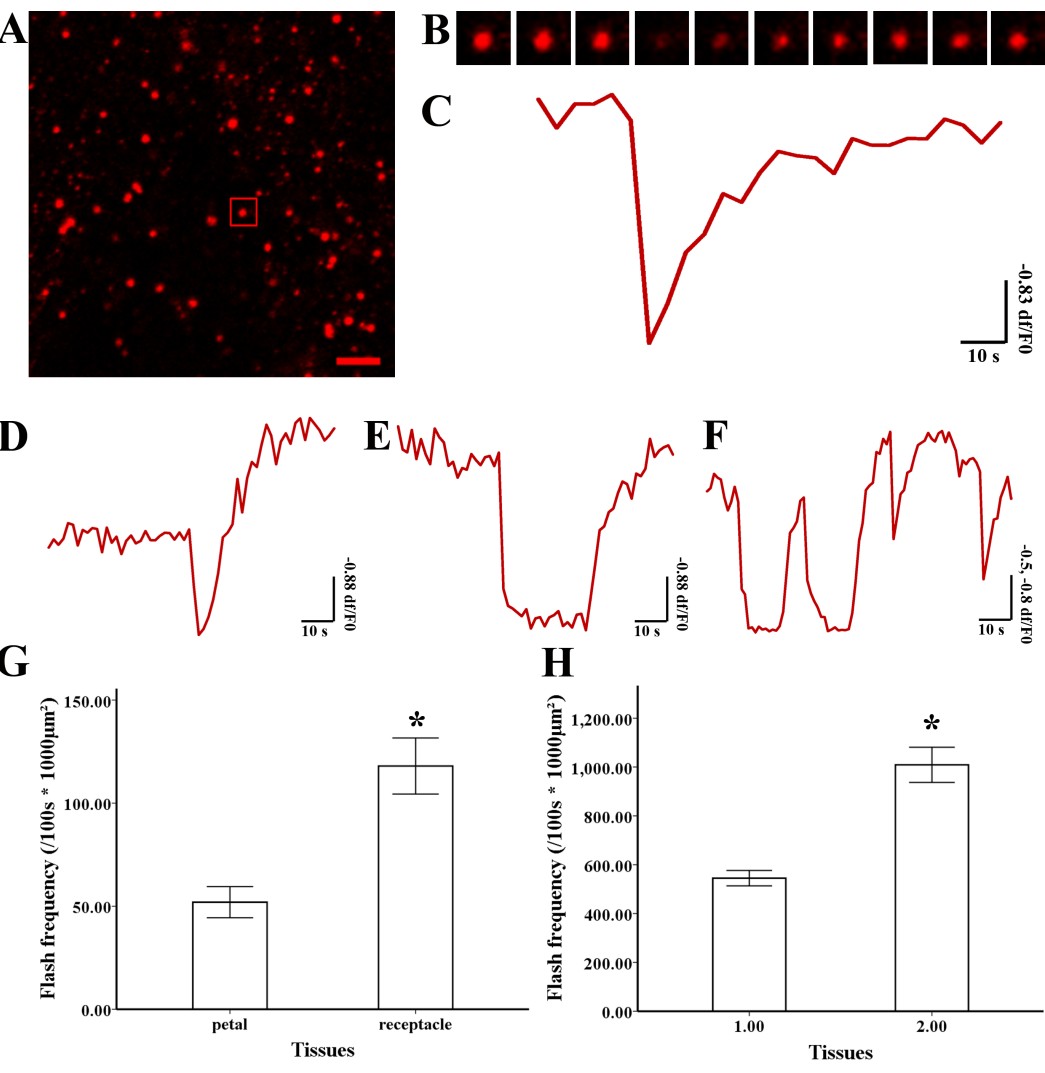

**Figure 5** **Depolarization of mitochondria membrane potential (ΔΨm) and frequency** (A) Isolated mitochondria labeled by TMRM. (B) Time-lapse images and (C) typical trace of depolarization of ΔΨm labeled by TMRM. Different types of trace of TMRM (D) instantaneous loss and recovery of ΔΨm, (E) instantaneous loss with the short period of stability before recovery of ΔΨm, (F) multi-event traces. Comparison of depolarization of ΔΨm frequency in mitochondria isolated from petal and style of *N. nucifera* (G). Comparison of depolarization of ΔΨm frequency in mitochondria isolated from petal and receptacle of *M. denudata* (H). Scale bar: 5 μm.

between the location of ROS and mitochondria in petals and pistils of two flower species (Fig. 2), indicating that ROS originate in mitochondria from floral tissues. ROS always act as signaling molecules to unlock the antioxidant system and maintain physiological activities in plants under salt stress (*Ahanger et al., 2017*). ROS are also involved in the energy-dissipating system that increases frost resistance in seedlings under freezing conditions (*Grabelnych et al., 2014*), and mitochondria contribute to ROS production through electron transfer from the respiratory chain in non-green tissues, such as flowers, and ROS homeostasis is regulated by the antioxidant system (*Rhoads et al., 2006*; *Rogers &*

*Munne-Bosch, 2016*). Considering the increased mitochondrial ROS production in pistils of magnolia and lotus in our study, mitochondrial ROS might be more involved in the respiratory metabolic signaling pathways in the pistils of these two flower species.

An increase in ROS accumulation can trigger ROS burst in plants (*Zandalinas & Mittler, 2018*), and basal mitochondrial ROS production is intimately linked with ROS flashes (*Wang et al., 2012*). The basal elevation of mitochondrial ROS triggers superoxide flashes (*Hou et al., 2013*). In our results, the simultaneous increase in ROS production and the frequency of superoxide flashes in the pistils indicated that increasing ROS production might trigger superoxide flashes. Superoxide flashes are involved in various stressful and pathophysiological conditions (*Fang et al., 2011*; *Wang et al., 2013*). Superoxide flashes are sensitive to mitochondrial respiration and a higher frequency of superoxide flash events acts as an early mitochondrial signal in response to physiological activities and oxidative stress (*Ma et al., 2011*; *Wei et al., 2011*). Superoxide flashes always respond to metabolic activities and act as a signal mediating disease (*Cao et al., 2013*; *Wang et al., 2013*). In addition, the frequency of superoxide flashes in early adulthood predicts the lifespan of an organism (*Shen et al., 2014*). Similarly, changes in superoxide flashes and fluorescence are closely related to respiratory activity in Arabidopsis and are affected by different respiratory substrates and inhibitors (*Schwarzlander et al., 2011*). Considering that superoxide flashes visualized by MitoSOX occurred at a similar frequency as cp-YFP-flashes in previous studies (*Wang et al., 2016a*; *Zhang et al., 2013*), our findings show that MitoSOX flashes in flower tissues reflect the nature of the flashes. An increase in the frequency of flashes in the pistil indicates that superoxide flashes together with mitochondrial ROS might be more involved in mitochondrial viability and physiological metabolism in flower pistils. Besides, superoxide flashes were strongly inhibited when electron transport was dysfunction (*Wang et al., 2008*). Superoxide flashes were markedly decreased by uncoupler of mitochondrial electron transport chain (*Zhang et al., 2013*). Also, the decrease of cp-YFP fluorescence caused by the addition of ADP and uncoupler FCCP was observed in mitochondria of Arabidopsis, which suggested the superoxide signal appeared to correlate with the magnitude of proton motive force (*Schwarzlander et al., 2011*). The decreasing of superoxide signal and flashes frequency in our study suggest that a strongly electrical transmembrane gradient is necessary for the production of superoxide flashes.

Spontaneous burst superoxide flashes are always consequential to the depolarization of mitochondrial $\Delta\Psi$ m (*Feng et al., 2017*). A cp-YFP flash is always accompanied by depolarization of the $\Delta\Psi$ m (*Li et al., 2012*). The global rise in mitochondrial basal ROS can trigger the depolarization of $\Delta\Psi$ m (*Zorov, Juhaszova & Sollott, 2014*) with further stimulation of ROS-induced ROS release resulting in an amplified ROS signal in response to oxidative challenge (*Kuznetsov et al., 2017*). Reversible variation of $\Delta\Psi$ m is associated with the release of ROS under different physiological conditions (*Kuznetsov et al., 2017*). The simultaneous change in the frequency of superoxide flashes and depolarization of the mitochondrial membrane potential in our study suggest that superoxide flashes are always accompanied by fluctuations in the $\Delta\Psi$ m. Although the biogenesis of superoxide flashes is closely related to depolarization of the $\Delta\Psi$ m, the genesis of the flashes is not only related to $\Delta\Psi$ m fluctuations. The incidence of $\Delta\Psi$ m fluctuations is higher than

that of superoxide flashes, because the cation and anion channels potentially contribute to fluctuation in the $\Delta\Psi$ m (*Wang et al., 2012*). Thus, the higher rate of depolarization of $\Delta\Psi$ m in our study suggests more ion exchange in mitochondria than the incidence of flashes.

## CONCLUSIONS

In conclusion, our study presents an efficient method to isolate functional mitochondria to study superoxide flashes. Superoxide flashes visualized by MitoSOX reflect the nature of the flash. Moreover, the simultaneous increase in MitoSOX flashes and depolarization of $\Delta\Psi$ m in mitochondria isolated from pistils demonstrate that mitochondria are involved in energy metabolism and physiological activities.

### Funding
This work was supported by the National Natural Science Foundation of China (No. J 1310002). The funders had no role in study design, data collection and analysis, decision to publish, or preparation of the manuscript.

### Grant Disclosures
The following grant information was disclosed by the authors:
National Natural Science Foundation of China: J 1310002.

### Competing Interests
The authors declare there are no competing interests.

### Author Contributions
- Chulan Zhang conceived and designed the experiments, performed the experiments, analyzed the data, contributed reagents/materials/analysis tools, prepared figures and/or tables, approved the final draft.
- Fengshuo Sun performed the experiments, analyzed the data, contributed reagents/materials/analysis tools, authored or reviewed drafts of the paper.
- Biao Xiong contributed reagents/materials/analysis tools, authored or reviewed drafts of the paper.
- Zhixiang Zhang conceived and designed the experiments, authored or reviewed drafts of the paper.

### Data Availability
The raw measurements are available in the Supplemental Files.

### Supplemental Information
Supplemental information for this article can be found online at http://dx.doi.org/10.7717/peerj.6708#supplemental-information.

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
