# Peer review of "Preparation of mitochondria to measure superoxide flashes in angiosperm flowers"

_PeerJ, doi:10.7717/peerj.6708_

## Round 0.1 · original submission · Major Revisions

Dear author

Your paper has been assessed by three reviewers and myself as academic Editor.

As you could see below , the manuscript needs a major revision.
The way to describe the experiments is not clear and inappropriate (are they performed on organs or isolated mitochondria?). Please remove the speculative conclusions about the evidence of mPTP opening. The two isolation techniques are quite interesting, so focus it more on these methods and less on the superoxide flashes, which are more debatable.

Reviewer 1 ·

Basic reporting

In general, criteria on basic reporting are met. However, see the general comments for the Author.

Experimental design

These standards are met.

Validity of the findings

Please see the general comments for the author (next section).

Additional comments

In this article, a fast method to prepare crude mitochondria from flowers is described. Mitochondria prepared with this method presented high phosphorylation activity; as assessed by its high respiration control ratios (RCR). In addition, they produced an electrical transmembrane potential when succinate was added as a respiratory substrate (observed with TMRM), which presented sudden decreases probably because the aperture of a permeability-transition pore (or other ion channels). Finally, superoxide flashes were observed in these respiring mitochondria.
My comments and suggested changes are:
1. A strong point is the method reported to prepare mitochondria. The authors appropriately point (in both the Introduction and the Discussion) that a quick preparation method often produces highly active organelles.
2. Another strong point is the measurement of superoxide flashes in a single mitochondrion. Using prepared mitochondria allows the observation of these flashes in media of known composition. The authors should take advantage of it. For instance, have superoxide flashes been observed in the presence of ADP + Pi? or in the presence of uncouplers? If these experiments have been done, state (and discuss) the results obtained. Results of experiments of this kind would strengthen the conclusions of this work (for instance, indicating if a strong electrical transmembrane gradient is necessary for the production of superoxide flashes).
3. The Figures and the Table illustrate well the findings reported. However, the information in headings is insufficient. For instance, it is necessary to state the method used to assess viability in Fig. 1. It is also convenient to describe briefly, in the same figure heading, the essentials of method B (e.g. “density gradient-purified mitochondria”.
4. In Fig. 1, much less “viable” (fluorescent) mitochondria are observed when the method B is used. In contrast, oxygen-consumption rates and RCR values are only slightly lower in mitochondria prepared with method B. How do you explain this discrepancy?
5. Some, mainly grammatical, suggested changes are indicated (as notes) directly on the PDF document.
Because of the above, I consider that this work can be accepted for publication once the suggested changes are followed.

Annotated reviews are not available for download in order to protect the identity of reviewers who chose to remain anonymous.

Reviewer 2 ·

Basic reporting

As presented, the article lacks of a clear focus. All along, authors mix the value of developing a new method for obtaining plant mithocondria, and using them for detecting ROS, with the physiological role of them. What is the goal?? reporting a new method or reporting the detection of ROS and its physiological relevance in tissue flower??

On the other hand, the number of references cited seems exceisive for the kind of article.

Experimental design

The experimental desing seem adecuate, however, it is not clearly presented. Authors should metion that they are following two methods in order to compare them and present a detailes description of each one.

Validity of the findings

See coments of the 1st section.

Additional comments

The MS present interesting and valuble data, however it is not clearly presented, since it emphasizes the development of new method in some sections whereas discussing the physyological relevance of ROS in floral tissues. These two elements, however are not properly articulated through the paper.

Reviewer 3 ·

Basic reporting

This manuscript describes a new and efficient technique to isolate crude mitochondria from flower elements. The quality of the isolated mitochondria appears to be quite high and the localization of these organelles and their elements
Nevertheless, the manuscript has two major faults that are: i) the results are described as the experiments were performed on intact tissues, not on isolated mitochondria. If this is not the case, the methods are missing this description; ii) the linkage between the loss in mitochondrial membrane potential and the manifestation of mitochondrial permeability transition is very speculative, since no further evidence is provided to establish the involvement of mPTP (e.g. by using cyclosporine A). Therefore, the title is quite misleading since the authirs showed the mitochondria preparation and the superoxide flashed but did not characterized the mitochondrial permeability transition, which requires a more complex characterization.
The manuscript in its present form needs major revisions.

Experimental design

Please, see above comments.

Validity of the findings

Please, see above comments.

---

## Round 0.2 · accepted · Accept

Dear authors

One reviewer has re-reviewed and confirmed that you have addressed all the requested corrections.

I congratulate you for the nice piece of work, which will add value to PeerJ.

# Reviewer 2 ·

Basic reporting

I am ready to recommend accepting this new version of the MS, since all observations previously made, were included.

Experimental design

No comments

Validity of the findings

No comments

Additional comments

I am ready to recommend accepting this new version of the MS, since all observations previously made, were included.